# Genetic variability and population structure of Ethiopian chickpea (*Cicer arietinum* L.) germplasm

Sintayehu Admas[1,2]*, Kassahun Tesfaye[2,3], Teklehaimanot Haileselassie[2], Eleni Shiferaw[1], K. Colton Flynn[4]

1 Ethiopian Biodiversity Institute, Addis Ababa, Ethiopia, 2 College of Natural Sciences, Addis Ababa University, Addis Ababa, Ethiopia, 3 Ethiopian Biotechnology Institute, Addis Ababa, Ethiopia, 4 Grassland Soil and Water Research Laboratory, USDA-ARS, Temple, Texas, United States of America

* sintayehu.admas@ebi.gov.et

**Data Availability Statement:** All relevant data are within the manuscript and its Supporting information files.

## Abstract

Evaluation of the genetic diversity and an understanding of the genetic structure and relationships of chickpea genotypes are valuable to design efficient germplasm conservation strategies and crop breeding programs. Information is limited, in these regards, for Ethiopian chickpea germplasms. Therefore, the present study was carried out to estimate the genetic diversity, population structure, and relationships of 152 chickpea genotypes using simple sequence repeats (SSR) markers. Twenty three SSR markers exhibited polymorphism producing a total of 133 alleles, with a mean of 5.8 alleles per locus. Analyses utilizing various genetic-based statistics included pairwise population Nei's genetic distance, heterozygosity, Shannon's information index, polymorphic information content, and percent polymorphism. These analyses exemplified the existence of high genetic variation within and among chickpea genotypes. The 152 genotypes were divided into two major clusters based on Nei's genetic distances. The exotic genotypes were grouped in one cluster exclusively showing that these genotypes are distinct to Ethiopian genotypes, while the patterns of clustering of Ethiopian chickpea genotypes based on their geographic region were not consistent because of the seed exchange across regions. Model-based population structure clustering identified two discrete populations. These finding provides useful insight for chickpea collections and *ex-situ* conservation and national breeding programs for widening the genetic base of chickpea.

## Introduction

Chickpea (*Cicer arietinum* L.) belongs to the family Fabaceae (formerly Leguminosae) and subfamily Faboideae. It is a diploid self-pollinated crop species having chromosome number of $2n = 2x = 16$ [1, 2] with a comparatively small genome size of 740 Mbp [3]. There are two main types of chickpeas, namely desi and kabuli, however, rarely pea-shaped chickpea types are available. Kabuli seed types (Macrosperma) are large, round or ram head, and cream-

**Funding:** The study is part of the first author PhD thesis funded by Ethiopia Biodiversity Institute and Addis Ababa University.

**Competing interests:** The authors declare there are no competing interests.

colored. It has medium to tall plant height, large leaflets and white flowers, and it contains no anthocyanin. The desi chickpea types (Microsperma) are characterized by pink, purple or blue flower color, darker seeds with a rough seed coat. It has small and angular seed shape. The plants are anthocyanin-rich and have tiny leaves and purplish blooms. Pea-shaped chickpea type is characterized by medium to small seed size and creamy color [4]. It may be a result of a cross between desi and kabuli types that has resulted in a sort of intermediate types [5].

Chickpea is believed to have originated from the South Eastern Turkey and the adjoining areas of Syria [2], where chickpea was domesticated and later spread to the secondary centers of diversity: north-east Africa, Mediterranean Europe and the Indian sub-continent and more recently to Mexico and Chile [6, 7]. India and Ethiopia have been proposed as secondary centers of diversity for cultivated chickpea [8]. Based on the presence of wild relative (*Cicer Cuneatum*) found in Northern Ethiopia [9] and the Archaeological evidence from Lalibela caves in Ethiopia with seed samples with seed sample dated at over 2500 years [10], Fikre et al. [11] suggested for reconsideration of Ethiopia as the origin of chickpea. However, with the current consensus, Ethiopia is considered as the secondary center of diversity for chickpea [8].

Chickpea is one of the first domesticated, cool season autogamous grain legume cultivated in more than 50 countries in subtropical and temperate regions throughout the world [12]. Chickpea is also one of the main crops in sub-Saharan Africa. Since 1961, its production area and productivity trend have been steadily increasing. Because, it serves as a cash crop, break crop for crop diseases managements, a rotational crop for soil fertility restoration, food for human beings, and feed for animals. It is also suitable for sustainable agriculture production system with little or no climatic shocks [11]. In Ethiopia, chickpea is the third largest food legume crop in sowing area and production and the second major export commodity next to white pea beans generating nearly 25% of the total legumes export earnings [13].

Knowledge of genetic diversity, structure and relationship among germplasm collections are vital to design appropriate germplasm conservation strategies and potential breeding programs [14]. These activities aid in the maintenance of highly diversified germplasm which provides ample opportunity to breeders to look for desirable traits for developing new and superior varieties. Comprehensive information on genetic diversity and structure can be generated from morphological, biochemical and molecular data. Ethiopian chickpea germplasm have been characterized extensively for phenotypic characteristics [15–20]. Although morphological markers allow the identification of genetic variation, it is masked by environmental factors and is minimized due to the lack of distinguishable morphological markers [21]. This deficiency in phenotypic characterization must be complemented with molecular methods that use molecular markers, which can generate reliable and reproducible information for the evaluation of diversity. In chickpea, various markers have been used for diversity analysis which includes microsatellite or simple sequence repeats (SSRs), sequence tagged microsatellite markers (STMS), expressed sequence tags (ESTs), single nucleotide polymorphism (SNP), cleaved amplified polymorphic sequences (CAPS), conserved intron spanning primers (CISP) and diversity arrays technology (DArT) markers [12].

Simple sequence repeats (SSRs) are short tandem repetitive DNA sequences with a repeat length of few (1–6) base pairs which are abundant, dispersed through the genome and are highly polymorphic in comparison with other molecular markers [22, 23]. SSRs have been the most widely used markers for genotyping chickpea because they are highly informative, co-dominant, high reproducible and transferable among related species, multi-allelic, and have high degree of polymorphism and extensive genome coverage [14, 24–26]. Moreover, SSR markers are three times as efficient as dominant markers for intraspecific analysis and are as efficient as other dominant markers in detecting interspecific variability [27]. In chickpea, large numbers of SSR markers have been characterized, identified and utilized extensively to

study genetic diversity and relationships to identify genetically diverse germplasm with beneficial traits for use in chickpea genome analysis, germplasm characterization, phylogenetic analysis and genetic diagnostics [14, 22, 25, 26]. The use of SSR markers for characterizing Ethiopian Chickpea has been implemented, however, the number of genotypes characterized so far [25, 28] were small in number as compared to the total number of genotypes conserved in the Ethiopian Biodiversity Institute (EBI) gene bank. The aim of this study was, therefore, to assess the patterns of genetic structure and the level of genetic diversity and relationships within and between Ethiopian Chickpea genotypes, improved chickpea varieties and breeding lines by using SSR markers.

## Materials and methods

### Plant materials

One-hundred fifty-two chickpea genotypes were considered for this study (Table 1). One-hundred thirty-eight are Ethiopian genotypes (landraces), eight are nationally released varieties from Ethiopian agricultural research centers and six were breeding lines accessed from the

Table 1. List of chickpea genotypes used for this study.

| Region | Zone | District | No # of genotypes | Name of genotypes |
|---|---|---|---|---|
| Amhara | East Gojjam1* | Awabel, Dejen, Enarj enawga, Enemay, Goncha siso enese, Guzamn | 22 | 228290, 240050, 207728-A, 207736-B, 30287-C, 30288-A, 30289-A, 30289-B, 30290-A, 30300-A, 41021-A, 41029-B, 41080-B, 41086-A, 41222-B, 41222-B, 41228-A, 41230-A, 41231-B, 41234-C, 41247-A and 41247-B |
| | East Gojjam2* | Debay telatgen, Hulet ej enese, Mota, Shebel berenta, Gonji, Yilmana densa, D/dare zuria, Dega damot, Adet | 23 | 207638, 212685-B, 212685-B, 215289-A, 30307-D, 30307-D, 30308-B, 30309-A, 30311-A, 30313-C, 30314-A, 30314-B, 30316-A, 41020-A, 41075-C, 41078-B, 41090-A, 41245-A, 41257-A, 41258-A, 41265-B, 41270-B and 41320-A |
| | North Gondar | Alefa, Belesa, Chilga, Dabat, Debark, Este, and Wegera | 16 | 207136-A, 207136-A, 207167-A, 207173-B, 207175-A, 207609-B, 207617-A, 225884-A, 227152-A, 227160-B, 227161-B, 24159-C, 241800-A, 241801-A, 41301-A and 9646-A |
| | Central Gondar | Gondar zuria, Kemkem, Mirab belesa, Mirab dendia, Misrak belesa | 14 | 30333, 207753-B, 236475-A, 30317-A, 30318-B, 30319-B, 30326-A, 30326-C, 30334-B, 30335-B, 30335-B, 30336-B, 30337-A and 41043-B |
| | North Shewa | Ankober, Debre Brehan, Efratana gidim, Mezezo Mojana, Mama lalo midir, Minjarna Shenkora, Siadebr and Tegulet | 22 | 207652, 215067-B, 215067-C, 235036-B, 235036-C, 237055-B, 30348-C, 41093-C, 41094-B, Dbarc-black 1, Dbarc-black 2, Dbarc-black 3, Dbarc-red 4, enewari1, enewari2, enewari3, enewari4, enewari5, enewari6, enewari7, tegulet1 and tegulet 2 |
| | North Wollo | Bugna, Dessie zuria, Guba lafto, Habru, Kelala, Kutaber, Sayint and Wereilu | 11 | 213050-B, 214732-A, 214734-C, 235032-A, 235032-B, 235034-C, 236194-A, 241804-C, 241804-D, 30347-B and 41116-A |
| Tigray** | | Rayaazebo, Medebay zana, Axum, Maychew and Wukro | 5 | 16586-A, 234050-B, 235391-A, 236459-B and 236467-A |
| Oromia | West Shewa | Ambo, Alem gena, Becho, Ejerie (addis alem), Jeldu and Kersana kondaltiiti | 12 | 207684, 207712, 207714, 207691-B, 207765-B, 207769-A, 209026-A, 228197-E, 41169-C, 41200-B, 41200-C and 41206-B |
| | Arsi Bale | Bekoji, Chole, Jeju, Goro, Robe, Robe market, and Girawa | 12 | 207670, 207664-A, 207679-B, 230796-C, 231331-A, 28741-A, 41035-C, 216854-C, 41136-A, 41030, 41034, and 41153-A |
| SNNP** | | Konso special | 1 | 225741-C |
| Exotic Genotypes | ICARDA Genotypes | | 6 | 125231***, 128699***, 140294***, 69757***, 70788*** and 9003*** |
| | Improved Varieties | | 8 | Dalota, Dhara***, Dubie, Ejere***, Mastewal, Minjar, Shahso*** and Teji*** |

*East Gojjam was grouped in two;

**Due to small number of genotypes, Tigray collections was grouped with North Wollo collections, Konso with Arsi Bale collections,

*** Kabuli types chickpea, while the rest are desi type chickpea

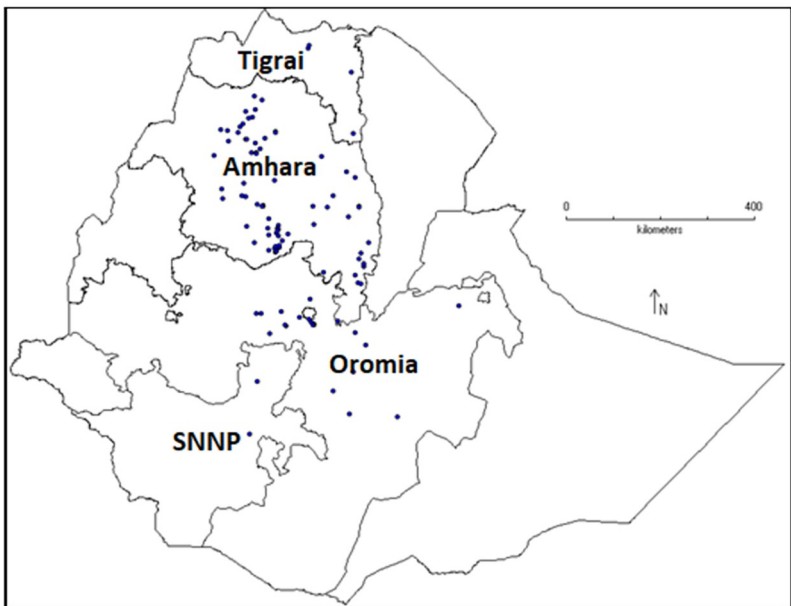

**Fig 1. Map showing the geographical distribution of Ethiopian chickpea germplasm.**

International Center for Agricultural Research in the Dry Areas (ICARDA). The geographical origin of the Ethiopian chickpea germplasm used in the study is indicated in Fig 1. Fig 1 was done using the software DIVA-GIS software [29] using the GPS coordinates of the collection sites (S1 Text). The genotypes were grown at Bakelo Research Station of Debre Brehan Agricultural Research Center in the 2018/2019 cropping seasons. Two weeks after planting, approximately equal amounts of bulk leaf samples were collected from five plants of each genotypes as suggested by Gilbert et al. [30] and then the leaves were stored in plastic Ziplock bags containing Silica gel.

## DNA extraction and quantification

Genomic DNA was extracted using the cetyltriethylammonium bromide (CTAB) technique [31] with slight adjustments. The leaf samples were ground into fine powder using pestle and mortar using 250 µl DNA extraction buffers (0.35M sorbitol, 0.1M Tris–HCl (pH 7.6), 0.005M EDTA, and 0.2M Tris-HCl, 0.05M EDTA, 2M NaCl and 2% CTAB, mixed in equal amounts). About 100 mg of ground leaf sample was transferred to 2 ml microcentrifuge tubes and 750 µl of extraction buffer was added. Tubes were maintained at 65 ˚C for one hour followed by chloroform-isoamyl alchohol (24:1) extraction. The DNA pellet was air dried and dissolved in 100 µl of 1× TE buffer. The quality and quantity of all DNA samples were checked using Nano Drop Spectrometer (ND-2000). In addition, agarose gel (0.8%) was used for checking the quality of the DNA by taking 30 genotypes selected systematically based on the result from the NanoDrop. The working DNA sample was diluted to obtain a final DNA concentration of 30–50 ng µL$^{-1}$.

## Polymerase chain reaction (PCR) and gel electrophoresis

Twenty-five SSR markers were used to carry out amplifications (Table 2). The SSR markers were purchased from Invitrogen Thermo Fisher from Scientific Life Technologies Europe BV, Nederlaenderna filial Sverige, Lindhagensgatan, Stockholm in 2020. The markers were selected

**Table 2. List and description of SSR primers used for finger printing 152 chickpea genotypes.**

| SSR Locus | Primer Sequence (5' to 3') | | Repeat motif | Size (bp) | Linkage Group | References |
|---|---|---|---|---|---|---|
| | Forward | Reverse | | | | |
| CaSTMS 11 | GTATCTACTTGTAATATTCTCTTCTCT | ATATCATAAACCCCCCAC | (GA)20 | 232 | 4 | [25] |
| CESSR 42 | TGGTTGAAGAAAAGAAGGTAGTG | CGGTTCACTAATGCAAAAACCT | (ACC)5 | 298 | | [24] |
| CESSR 62 | ACCAGCTGCTAGACCTGATGTT | GCAATAAAACAAAATCCTCACACC | (TGA)5, (TAT)3 | 245 | | [24] |
| CESSR 71 | TTGTAGTTCTCCTCTCTCTCTCTC | CATCAAAACCAAACCTATGGAG | (CT)C(CT)8, (CT)6, (CT)11 | 295 | | [24] |
| CESSRDB 45 | AGATGGTTTGAATGTTGAGG | CACTTGACCCTTTGATTGTT | (AT)7(AG)5 | 295 | | [24] |
| CESSRDB 54 | AGTGTTGTGGGTTTCATTTC | TTGATTTGCCAAAGTACACA | (TTA)5 | 221 | | [24] |
| GA 11 | GTTGAGCAACAAAGCCACAA | TTCTTGTCTGGTTGTGTGAGC | (CT)21 | 159 | 3, 1, 2, 6 | [25, 40] |
| GA 24 | TTGCCAAAACCAATAACTCTG | TCCCTTTTACACAAGGCCAG | (GA)19 | 203 | 1,2,4 | [25, 40] |
| GA-20 | TATGCACCACACCTCGTACC | TGACGGAATTCGTGATGTGT | (CT)23 | 174 | 2,6 | [38, 42] |
| NCPGR 100 | CCATTTTCTACAATCTCATGTCT | GTAGAAAGAGCCAAGAGGCA | CT)15N42(CT)2CC(CT)5TT (CT)6AT(CT) 7 | 263 | 1 | [25] |
| NCPGR 45 | TGTTTTCAAATCAAACAGGC | GATACACACCAAGGCACAGT | (CT)2GTCAT(CT)5CC(CT)2CC (CT)17 | 223 | 2 | [25] |
| NCPGR 53 | CCCTCCTTCTTGCTTACAAA | TAATGGTGAACGAATCATGG | (CT)5CA(CT)CA(CT)10CA(CT) 4CA(CT)TA(CT)4GTCA(CT)12 | 194 | 1 | [25] |
| NCPGR 94 | GGTTTGATGTGTTCTTGGCT | CCCTCAATTCCCTCGATTTA | (CT)25 | 176 | 5 | [25] |
| SSR 1 | TGAATTTTGTTTACCACCCCTC | TTTGGCTTATTCTGTTCTTCCC | (AG)20 | 157 | | [33] |
| SSR 22 | GCTTTCCCTTTACTTCTTGGGT | TGCTATTCAAGTCTCCCTCCTC | (AATG)5 | 275 | | [33] |
| SSR 31 | TAACGACAACGACAACAACAGC | GCCATTCCAGAGAGCCTTG | (AAC)14 | 161 | | [33] |
| SSR 4 | GACAAAACAACCTCCCAAGAAA | AACAACGACAACAACAACAACG | (TTG)6 | 279 | | [33] |
| SSR 5 | GAGCCCTGAAATGAAGAAAGAA | CACCTTTGAGCCCTAGTCTGTT | (AAAT)5 | 387 | | [33] |
| SSR 60 | GGTCATGTTGATTTCTCACCAA | GAACTTTCCGCACACGTTATG | (AAAT)6 | 337 | | [33] |
| TA 144 | ATTTTAATCCGGTGAATATTACCTTT | GTGGAGTCACTATCAACAATCATACAT | (TAA)27 | 241 | 5,6,8 | [25, 40] |
| TA 18 | AAATAATCTCCACTTCACAAATTTTC | ATAAGTGCGTTATTAGTTTGGTCTTGT | (TAA)24 | 147 | 7,5,6 | [25, 40, 42] |
| TA 76s | TCCTCTTCTTCGATATCATCA | CCATTCTATCTTTGGTGCTT | (AAT)7(AAT)4[ACT(AAT)11] 2ACT(AAT)3TAT(AAT)2(ATT) 5 | 206 | 3,4 | [25, 40] |
| TR 1 | CGTATGATTTTGCCGTCTAT | ACCTCAAGTTCTCCGAAAGT | (TAA)31 | 224 | 5,6 | [25, 40] |
| TR 2 | GGCTTAGAGTTCAAAGAGAGAA | AACCAAGATTGGAAGTTGTG | (TAA)36 | 210 | 3 | [25, 40] |
| TR 29 | GCCCACTGAAAAATAAAAAG | ATTTGAACCTCAAGTTCTCG | (TAA)8TAGTAATAG(TAA)32 | 197–251 | 7,5,1,3 | [25, 40, 42] |

based on polymorphic information content (PIC), allelic richness and herozygosity reports from various scientists [14, 24–26, 32, 33]. These SSR markers were developed from sequence information obtained by various authors [4, 24, 33–40]. The description of the primers is indicated in Table 2.

PCR reaction was performed with a Hybaid PCR express thermal cycler (Hybaid, UK) after optimizing the amplification conditions for each primer pair in a total volume of 10µl containing 50 ng of DNA,1.5mM MgCl$_2$, 0.2mM dNTPs, 0.4 mM each of the forward and reverse primers and 0.05U/µlt Taq polymerase. The PCR was programmed at an initial denaturation step of 3 min at 94 ˚C followed by 35 cycles of 20 s denaturation at 94 ˚C, annealing at 55 to 60˚C (depending on the primer) for 50 s, initial extension at 72˚C for 50 s, and final extension at 72˚C for seven mins. Before determining polyacrylamide gel staining, the amplified products were checked for the reproducibility of PCR products using a 2% agarose gel stained with ethidium bromide in a TBE buffer and were visualized on a UVITEC gel doc (UVITEC, UK).

The resolution of PCR products was done on 6% polyacrylamide gel in 0.5x TBE buffer with a 6x DNA loading dye. Electrophoresis was carried out on a vertical electrophoresis set up using a standard DNA ladder (100 bp, Solis Biodyne, Estonia). The vertical electrophoresis was run with 100V for two hrs and 30mins, and stained using silver staining developed by Huang et al. [41]. Then gel pictures were taken using digital camera. The band sizes were determined using UVITEC (UVITEC, Cambridge, UK) software. Primer bands that were unclear or absent were sorted and repeated. Non-polymorphic, missing, faint and distorted gels were disregarded at scoring and only records of 23 primers with clear polymorphic bands were considered for statistical analysis.

## Scoring SSR data and statistical analysis

Allelic data were recorded for each of the microsatellites markers for each genotype with the help of UVITEC software as well as visually. The allelic data scores locus-based diversity indices including the number of alleles (Na), effective number of alleles (Ne), observed heterozygosity (Ho), expected heterozygosity (He), Shannon's information index (I), number of privet alleles (NPA), fixation index, percent polymorphism and unique alleles were recorded using GenAlEx v.6.502 [43]. Estimates of genetic differentiation were computed by analysis of molecular variance (AMOVA) to partition total genetic variation into within and among population subgroups using GenAlEx 6.502 [43]. PowerMarker 3.25 [44] was used to estimate major allele frequency (MAF), Gene Diversity (GD), and polymorphic information content (PIC).

The allelic data scored was used to analyze principal coordinate analysis (PCoA) using GenAlEx v.6.502 [43]. Dendrogram tree was constructed based on Nei's genetic distance using PowerMarker 3.25 and the tree was visualized using Molecular Evolutionary Genetic Analysis (MEGA 6) [45]. Dendrogram was constructed by the unweighted pair-group method with arithmetic averages (UPGMA) [46]. The structure of the population was analyzed based on the Bayesian model-based clustering method using Structure 2.3.4 software as suggested by Pritchard et al., [47]. This software assumes a model in which there are K populations, which contribute to the genotype of each individual. Burning period of 50,000 and 100,000 Markov Chain Monte Carlo (MCMC) iterations were used with independent replications of 10 times for each K value (K = 1 to 10) assuming an admixture model and uncorrelated allele frequencies. A web-based Structure Harvester program [48] was employed to determine the most likely value of K for each test [49].

## Results

### Microsatellite repeats locus diversity

The polyacrylamide gel electrophoresis pictures and the estimated genetic diversity parameters of the SSR locus diversity are indicated in Fig 2 and Table 3, respectively. The Visual observations on the gels of the amplification products of the respective markers revealed the existence of low (Fig 2C and 2D) to high (Fig 2A and 2B) level of polymorphism in the Ethiopian genotypes and exotic genotypes depending on the types of primer used. Among 152 chickpea genotypes a total of 133 alleles with an average value of 5.8 alleles per SSR were recorded. The allelic richness (Na) per locus varied widely among markers, ranging from two (CESSRDB 45, SSR22, and SSR 5) to 16 (TR 1). The number of effective alleles (Ne) ranged between 1.3 (CESSRDB 45) and 7.6 (TR 29), with an overall mean of 3.2. Shannon's information index (I) was ranged from 0.4 (CESSRDB 45) to 2.1 (TR 1 and TR 29) with mean of 1.2. The average observed heterozygosity (0.4) was lower than the expected heterozygosity (0.6) and unbiased expected heterozygosity (0.6). The inbreeding coefficient (Fis) and fixation index (Fit) values

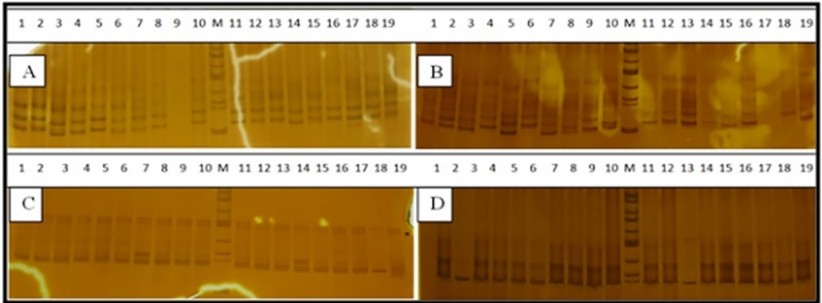

**Fig 2. Polyacrylamide gel electrophoresis pictures obtained with TR 29 (A), TR 1 (B), SSR 22 (C) and CESSR 42 (D) primers detected in chickpea genotypes.** The lane numbers identify serial no of genotypes and M stands for 100 bp DNA ladder.

**Table 3. Estimated genetic diversity parameters of 23 SSR markers in 152 chickpea genotypes.**

| Locus | Na | Ne | I | Ho | He | uHe | F | Ht | Fis | Fit | Fst | Nm | MAF | PIC | NPA | GD |
|---|---|---|---|---|---|---|---|---|---|---|---|---|---|---|---|---|
| CaSTMS 11 | 5 | 2.0 | 0.8 | 0.0 | 0.4 | 0.5 | 1.0 | 0.7 | 1.0 | 1.0 | 0.4 | 0.5 | 0.49 | 0.58 | 0 | 0.63 |
| CESSR 42 | 5 | 2.7 | 1.1 | 0.0 | 0.6 | 0.6 | 1.0 | 0.8 | 1.0 | 1.0 | 0.2 | 1.1 | 0.32 | 0.71 | 0 | 0.75 |
| CESSR 62 | 3 | 1.9 | 0.7 | 0.5 | 0.4 | 0.4 | -0.1 | 0.5 | -0.2 | 0.1 | 0.2 | 0.9 | 0.59 | 0.47 | 0 | 0.55 |
| CESSR 71 | 5 | 2.8 | 1.2 | 0.4 | 0.6 | 0.7 | 0.4 | 0.7 | 0.4 | 0.4 | 0.1 | 1.7 | 0.44 | 0.67 | 0 | 0.71 |
| CESSRDB 45 | 2 | 1.3 | 0.4 | 0.5 | 0.2 | 0.2 | -1.4 | 0.2 | -1.1 | -1.0 | 0.0 | 6.0 | 0.88 | 0.19 | 0 | 0.22 |
| CESSRDB 54 | 4 | 1.8 | 0.7 | 0.2 | 0.4 | 0.4 | 0.6 | 0.6 | 0.6 | 0.7 | 0.3 | 0.7 | 0.53 | 0.48 | 0 | 0.56 |
| GA 11 | 6 | 2.8 | 1.2 | 0.3 | 0.6 | 0.6 | 0.6 | 0.7 | 0.5 | 0.6 | 0.1 | 2.0 | 0.42 | 0.66 | 0 | 0.70 |
| GA 24 | 6 | 2.6 | 1.2 | 0.3 | 0.6 | 0.6 | 0.6 | 0.7 | 0.6 | 0.7 | 0.2 | 1.3 | 0.39 | 0.67 | 0 | 0.71 |
| GA-20 | 6 | 3.1 | 1.3 | 0.5 | 0.7 | 0.7 | 0.3 | 0.8 | 0.3 | 0.4 | 0.1 | 1.5 | 0.38 | 0.73 | 0 | 0.76 |
| NCPGR 100 | 3 | 1.9 | 0.7 | 0.5 | 0.5 | 0.5 | 0.0 | 0.5 | 0.0 | 0.1 | 0.1 | 3.1 | 0.56 | 0.38 | 1 | 0.49 |
| NCPGR 45 | 4 | 2.6 | 1.1 | 0.7 | 0.6 | 0.6 | -0.2 | 0.7 | -0.2 | -0.1 | 0.1 | 2.6 | 0.45 | 0.62 | 0 | 0.67 |
| NCPGR 53 | 4 | 2.2 | 0.9 | 0.4 | 0.5 | 0.5 | 0.3 | 0.7 | 0.3 | 0.4 | 0.2 | 0.9 | 0.41 | 0.59 | 0 | 0.65 |
| NCPGR 94 | 10 | 3.9 | 1.5 | 0.6 | 0.7 | 0.8 | 0.2 | 0.8 | 0.2 | 0.3 | 0.1 | 1.9 | 0.26 | 0.80 | 1 | 0.82 |
| SSR 1 | 6 | 3.2 | 1.2 | 0.5 | 0.7 | 0.7 | 0.2 | 0.7 | 0.2 | 0.3 | 0.1 | 3.3 | 0.44 | 0.66 | 1 | 0.70 |
| SSR 22 | 2 | 1.3 | 0.4 | 0.3 | 0.2 | 0.2 | -0.2 | 0.2 | -0.2 | -0.2 | 0.0 | 7.4 | 0.85 | 0.22 | 0 | 0.25 |
| SSR 4 | 6 | 4.0 | 1.5 | 0.6 | 0.7 | 0.8 | 0.2 | 0.8 | 0.2 | 0.2 | 0.1 | 2.9 | 0.29 | 0.78 | 0 | 0.80 |
| SSR 5 | 2 | 1.8 | 0.6 | 0.0 | 0.4 | 0.4 | 1.0 | 0.5 | 1.0 | 1.0 | 0.2 | 1.4 | 0.55 | 0.37 | 0 | 0.49 |
| SSR 60 | 6 | 3.1 | 1.3 | 0.4 | 0.7 | 0.7 | 0.4 | 0.8 | 0.5 | 0.5 | 0.1 | 1.8 | 0.33 | 0.72 | 0 | 0.75 |
| TA 144 | 3 | 2.4 | 0.9 | 0.0 | 0.6 | 0.6 | 1.0 | 0.7 | 1.0 | 1.0 | 0.2 | 1.4 | 0.35 | 0.59 | 0 | 0.66 |
| TA 18 | 8 | 5.2 | 1.8 | 0.6 | 0.8 | 0.8 | 0.3 | 0.9 | 0.3 | 0.3 | 0.1 | 3.6 | 0.20 | 0.84 | 0 | 0.85 |
| TR 1 | 16 | 7.3 | 2.1 | 1.0 | 0.8 | 0.9 | -0.2 | 0.9 | -0.2 | -0.1 | 0.1 | 3.7 | 0.14 | 0.90 | 2 | 0.90 |
| TR 2 | 9 | 6.0 | 1.9 | 1.0 | 0.8 | 0.9 | -0.2 | 0.9 | -0.2 | -0.2 | 0.1 | 4.7 | 0.22 | 0.86 | 0 | 0.87 |
| TR 29 | 12 | 7.6 | 2.1 | 1.0 | 0.9 | 0.9 | -0.2 | 0.9 | -0.2 | -0.1 | 0.1 | 4.5 | 0.11 | 0.90 | 0 | 0.91 |
| Total | 133 | 73.3 | - | - | - | - | - | - | - | - | - | - | - | - | 5 | - |
| Mean | 5.8 | 3.2 | 1.2 | 0.4 | 0.6 | 0.6 | 0.2 | 0.7 | 0.3 | 0.3 | 0.1 | 2.6 | 0.42 | 0.63 | - | 0.67 |
| SE | 0.2 | 0.1 | 0.0 | 0.0 | 0.0 | 0.0 | 0.0 | 0.2 | 0.1 | 0.1 | 0.0 | 0.4 | - | - | - | - |

**Key**: Na = number of alleles detected per locus; Ne = number of effective alleles; I = Shannon's Information Index; Ho = Observed Heterozygosity; He = Expected Heterozygosity; uHe = Unbiased Expected Heterozygosity; F = fixation Index; Ht = Total Expected Heterozygosity; Fis = inbreeding coefficient; Fit = fixation index; Fst = genetic differentiation; Nm = Gene flow; MAF = major allele frequency; PIC = Polymorphic Information Center; NPA = number of privet alleles; GD = Gene Diversity; and SE is standard error.

**Table 4. Summary of parameters for genetic diversity in chickpea population from different geographic origins.**

| Populations | Population diversity parameters | | | | | | | | | | |
|---|---|---|---|---|---|---|---|---|---|---|---|
| | Na | Ne | I | Ho | He | UHe | F | % P | NPA | MAF | PIC |
| East Gojjam1 | 4.7 | 3.3 | 1.2 | 0.5 | 0.6 | 0.6 | 0.2 | 95.7 | 0.00 | 0.50 | 0.55 |
| East Gojjam2 | 5.3 | 3.6 | 1.3 | 0.5 | 0.6 | 0.6 | 0.2 | 100.0 | 0.04 | 0.49 | 0.58 |
| North Gondar | 4.7 | 3.2 | 1.2 | 0.4 | 0.6 | 0.6 | 0.3 | 100.0 | 0.00 | 0.52 | 0.54 |
| Central Gondar | 4.6 | 3.0 | 1.1 | 0.4 | 0.6 | 0.6 | 0.3 | 100.0 | 0.00 | 0.55 | 0.51 |
| North Shewa | 5.1 | 3.6 | 1.2 | 0.4 | 0.6 | 0.6 | 0.3 | 100.0 | 0.09 | 0.49 | 0.57 |
| North Wollo | 4.9 | 3.2 | 1.2 | 0.4 | 0.6 | 0.6 | 0.3 | 100.0 | 0.04 | 0.52 | 0.55 |
| West Shewa | 4.4 | 3.1 | 1.1 | 0.4 | 0.6 | 0.6 | 0.3 | 100.0 | 0.00 | 0.52 | 0.54 |
| Arsi Bale | 4.6 | 2.9 | 1.1 | 0.5 | 0.6 | 0.6 | 0.2 | 100.0 | 0.04 | 0.56 | 0.52 |
| Exotic Genotypes | 3.7 | 2.7 | 1.0 | 0.5 | 0.5 | 0.6 | 0.0 | 100.0 | 0.00 | 0.54 | 0.49 |
| Mean | 4.7 | 3.2 | 1.1 | 0.4 | 0.6 | 0.6 | 0.2 | 99.5 | 0.22 | - | - |
| SE | 0.2 | 0.1 | 0.0 | 0.0 | 0.0 | 0.0 | 0.0 | 0.5 | - | - | - |

Key: Na = number of alleles detected per locus; Ne = number of effective alleles; I = Shannon's Information Index; Ho = Observed Heterozygosity; He = Expected Heterozygosity; uHe = Unbiased Expected Heterozygosity; F = fixation Index; % P = percent polymorphism; NPA = number of private Alleles; MAF = major allele frequency; PIC = Polymorphic Information Center and SE = standard error.

ranged from -1.1 to 1.0 and -1.0 to 1.0, respectively. The major allele frequency varied from 0.11 (TR 29) to 0.88 (CESSRDB 45) with an average of 0.49. Polymorphic information content (PIC) values ranged from 0.19 (CESSRDB 45) to 0.9 (TR 1 and TR 29) with an average of 0.58. Seventeen markers (73.1%) had a PIC score of 0.5 and above. Gene diversity values ranged from 0.22 (CESSRDB 45) to 0.91 (TR 29) with an average of 0.67. Out of the total number of alleles only five alleles (0.07% of the total alleles detected) were private alleles observed in locus NCPGR 100 in genotypes 30307-A from East Gojjam2, TR 1 in genotypes Enewari1 from North Shewa, SSR 1 in genotypes enewari1 from North Shewa, SSR1 in genotypes 30347-B from North Wollo and NCPGR 94 in genotypes 41030 from Arsi Bale.

## Genetic diversity in chickpea genotypes and population

The genetic diversity indices for chickpea genotypes based on geographic origins is summarized in Table 4. The observed numbers of alleles (Na) were in the range of 3.7 (Exotic Genotypes) to 5.3 (East Gojjam2). The number of effective alleles (Ne) ranged from 2.7 (Exotic Genotypes) to 3.6 (East Gojjam2 and North Shewa). Shannon's information index (I) ranged from 1.0 (Exotic Genotypes) to 1.3 (East Gojjam 2). The mean of the observed heterozygosity (0.4) is less than expected heterozygosity (0.6) and unbiased expected heterozygosity (0.6). The inbreeding coefficient (F) estimate ranged from 0.0 (Exotic Genotypes) to 0.3 (North Gondar, Central Gondar, North Shewa, North Wollo and West Shewa) with the average of 0.2. The mean percentage of polymorphic locus (% P) across population was 99.5% varying from 95.7 to 100%. Higher values of number of private alleles (NPA) were observed in populations of East Gojjam 2 (0.04), North Shewa (0.09), North Wollo (0.04) and Arsi-Bale (0.04).

## Analysis of molecular variance (AMOVA) and partitioning genetic diversity

The AMOVA showed that 88% of the allelic variation was attributed to individual genotypes within populations, while only 12% was distributed among populations (Table 5). The local population contributed 7% (West Shewa) to 14% (East Gojjam2), while the exotic genotypes contributed 7.6% of the total variation. The value of pairwise comparisons of population

**Table 5. Analysis of Molecular Variance (AMOVA) showing the distribution of genetic diversity within and among populations of chickpea genotypes from different sources of origins.**

| Source of variations | Degree of freedom | Sum square | Mean square | Variance Estimated variances | Proportion of explained variance in % | Statistics | Value | P value |
|---|---|---|---|---|---|---|---|---|
| **Among Pops** | 8 | 288.993 | 35.874 | 0.856 | 12 | | | |
| **Within Pops** | 295 | 2114.224 | 7.167 | 7.167 | 88 | PhiPT | 0.107 | 0.001 |
| *East Gojjam1* | | 306.500 | | | 12.8 | | | |
| *East Gojjam2* | | 336.413 | | | 14 | | | |
| *North Gondar* | | 221.688 | | | 9.2 | | | |
| *Central Gondar* | | 186.607 | | | 7.8 | | | |
| *North Shewa* | | 317.682 | | | 13.2 | | | |
| *North Wollo* | | 219.313 | | | 9.1 | | | |
| *West Shewa* | | 167.500 | | | 7.0 | | | |
| *Arsi Bale* | | 177.308 | | | 7.4 | | | |
| *Exotic Genotypes* | | 181.214 | | | 7.6 | | | |
| **Total** | 303 | 2402.217 | | 8.023 | 100 | | | |

differentiation (Fst) and Gene flow (Nm) among geographical regions of chickpea populations are indicated in Table 6. Highest Fst value was observed between exotic genotypes and chickpea populations from central Gondar (0.18) and Arsi-Bale (0.18), while the lowest was recorded between chickpea population of North Shewa and Central Gondar (0.05), North Wollo (0.05) versus North Shewa, and Arsi-Bale and West Shewa (0.05). Generally the exotic genotypes showed high Fst value compared to chickpea genotypes of Ethiopian origin than pairwise comparison between chickpea population within Ethiopian origin. Gene flow (Nm) between and within geographical regions was recorded from 1.16 (Arsi-Bale versus Exotic Genotypes) to 3.96 (East Gojjam2 versus North Gondar).

## Principal coordinates analysis (PCoA)

The multivariate principal coordinate analysis (PCA) of the molecular data showed that the first 3 coordinates were important and accounted for 26.6% of the variation; PCs 1 (14.0%), 2 (6.9%), and 3 (5.7%). The PCA plots of PC 1 versus PC 2 using factorial analysis of GenAlEx showed the exotic genotypes were clustered in quadrant I entirely, while a wide dispersion of Ethiopian genotypes across the four quadrants (Fig 3) were observed without considering their geographic origin. Genotypes collected from East Gojjam1 clustered in quadrant III (eight

**Table 6. Pairwise population differentiation (Fst) values above diagonal and gene flow (Nm) below diagonal among chickpea populations from different growing geographic areas based on the probability level based on 999 permutations.**

| Population | East Gojjam1 | East Gojjam2 | North Gondar | Central Gondar | North Shewa | North Wollo | West Shewa | Arsi Bale | Exotic Genotypes |
|---|---|---|---|---|---|---|---|---|---|
| **East Gojjam1** | 0 | 0.07* | 0.08* | 0.13** | 0.10** | 0.14** | 0.13** | 0.11** | 0.16** |
| **East Gojjam2** | 3.14 | 0 | 0.06* | 0.09* | 0.07** | 0.14** | 0.10** | 0.09** | 0.14** |
| **North Gondar** | 2.90 | 3.96 | 0 | 0.13** | 0.07** | 0.12** | 0.11** | 0.13** | 0.12** |
| **Central Gondar** | 1.69 | 2.57 | 1.74 | 0 | 0.05** | 0.10** | 0.09** | 0.13** | 0.18** |
| **North Shewa** | 2.25 | 3.24 | 3.11 | 4.87 | 0 | 0.05** | 0.08** | 0.10** | 0.16** |
| **North Wollo** | 1.57 | 1.58 | 1.80 | 2.19 | 4.67 | 0 | 0.08** | 0.11** | 0.16** |
| **West Shewa** | 1.66 | 2.37 | 2.12 | 2.63 | 2.96 | 2.79 | 0 | 0.05** | 0.15** |
| **Arsi/Bale** | 2.05 | 2.59 | 1.73 | 1.75 | 2.26 | 2.03 | 4.35 | 0 | 0.18** |
| **Exotic Genotypes** | 1.29 | 1.59 | 1.75 | 1.12 | 1.32 | 1.33 | 1.47 | 1.16 | 0 |

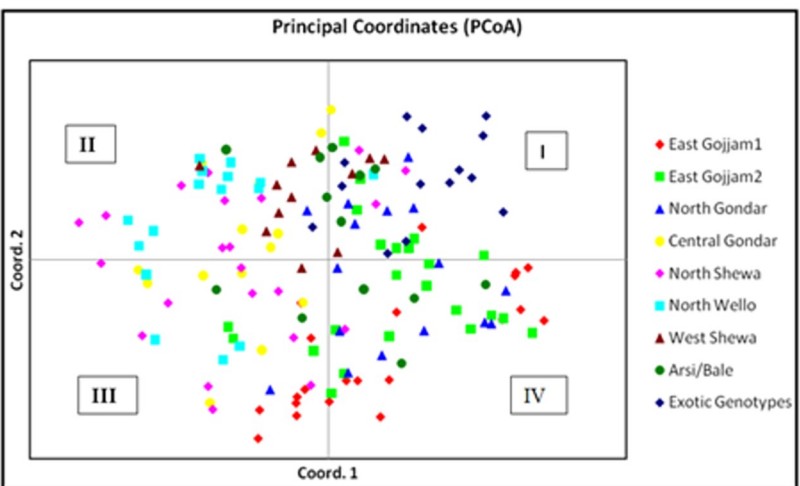

**Fig 3. Two-dimensional plot obtained from principal coordinate analysis (PCoA) of 152 chickpea accessions using 23 SSR markers.**

genotypes), and IV (13 genotypes) forming small sub-clusters in both quadrants. A single genotype from this zone falls in quadrant I. East Gojjam2 collections clustered in quadrant I (8 genotypes), III (3 genotypes), and IV (12 genotypes) sowing a tendency of forming sub-clusters in each quadrant. Genotypes of North Gondar clustered in quadrant I (5 genotypes) and IV (9 genotypes). The remaining two genotypes were grouped in cluster II and III. The majority of the genotypes collected from Central Gondar were clustered in quadrant III (8 genotypes). The remaining five genotypes and one genotype fall in quadrant II and I, respectively. Genotypes of North Shewa collection were clustered in quadrant I (3 genotypes), II (7 genotypes), III (10 genotypes), and IV (1 genotype). Genotypes from North Wollo formed two sub clusters in quadrant II (12 genotypes). The remaining one and four genotypes clustered in quadrant I and III, respectively. Genotypes from West Shewa were clustered in quadrant I (4 genotypes) and II (8 genotypes). Genotypes of Arsi Bale appeared be widely distributed in all quadrants, I (4 genotypes), II (3 genotypes), III (2 genotypes), and IV (4 genotypes).

## Genetic distance

The pairwise Nei's unbiased genetic distances (above diagonal) and unbiased genetic identity values (below diagonal) for all the chickpea populations representing the growing regions are shown in Table 7. The matrix of pairwise Nei's unbiased genetic distances between populations showed a close genetic distance between North Shewa and Central Gondar populations (0.09), North Wollo and North Shewa (0.09), and West Shewa and Arsi-Bale (0.09). On the other hand, the largest genetic distance (0.37) was obtained between population of Arsi-Bale and exotic genotypes. Generally, genetic distances between Ethiopian chickpea population and exotic genotypes were greater than any other combinations of paired populations within Ethiopia. The highest genetic identity value (0.92) was recorded between North Shewa population and Central Gondar population and the lowest genetic identity value (0.68) was recorded between Arsi-Bale and exotic genotypes. The genetic identity pairwise comparisons within genotypes of Ethiopian origins were more than the comparison between exotic with genotypes of Ethiopian origins.

**Table 7. Pairwise Population Matrix of Nei's unbiased genetic distance (DA) above diagonal and Pairwise Population Matrix of Nei unbiased genetic identity below diagonal among chickpea populations from different origins.**

| Populations | East Gojjam1 | East Gojjam2 | North Gondar | Central Gondar | North Shewa | North Wollo | West Shewa | Arsi-Bale | Exotic Genotypes |
|---|---|---|---|---|---|---|---|---|---|
| East Gojjam1 | * | 0.15 | 0.15 | 0.23 | 0.21 | 0.25 | 0.25 | 0.21 | 0.34 |
| East Gojjam2 | 0.86 | * | 0.11 | 0.17 | 0.15 | 0.31 | 0.20 | 0.17 | 0.29 |
| North Gondar | 0.86 | 0.89 | * | 0.22 | 0.14 | 0.25 | 0.21 | 0.25 | 0.26 |
| Central Gondar | 0.77 | 0.84 | 0.78 | * | 0.09 | 0.19 | 0.15 | 0.24 | 0.38 |
| North Shewa | 0.81 | 0.86 | 0.87 | 0.92 | * | 0.09 | 0.16 | 0.20 | 0.35 |
| North Wollo | 0.76 | 0.76 | 0.78 | 0.83 | 0.91 | * | 0.16 | 0.20 | 0.33 |
| West Shewa | 0.76 | 0.82 | 0.81 | 0.86 | 0.85 | 0.86 | * | 0.09 | 0.29 |
| Arsi-Bale | 0.81 | 0.84 | 0.78 | 0.79 | 0.82 | 0.82 | 0.91 | * | 0.37 |
| Exotic Genotypes | 0.71 | 0.75 | 0.75 | 0.69 | 0.71 | 0.72 | 0.75 | 0.68 | * |

## Cluster analysis

A dendrogram tree based on Nei's genetic distances was constructed using PowerMarker V3.25 software. The result from UPGMA based dendrogram shows that nine chickpea populations from different geographic origins were grouped into two major clusters (Fig 4). The first cluster contained the exotic genotype population, while cluster II consisted of the Ethiopian populations. Cluster II was divided into three sub-clusters showing the tendencies of grouping the neighboring regions together. The 152 genotypes were divided into two major clusters (Fig 5). Cluster I had 14 genotypes which were exclusively from the exotic genotypes. Cluster II was further sub divided into six distinct sub-clusters with variable number of genotypes in each sub-cluster. Sub-cluster 1 consisted of 27 genotypes with the following proportions, 22 (81.5%) from East Gojjam 1, two (7.4%) from East Gojjam 2, and three (11.1%) from Arsi-Bale. Sub-cluster 2 was composed of 25 genotypes of which 19 (76%) genotypes were from East Gojjam 2 and six (24%) genotypes from North Gondar. Sub-cluster 3 was composed of 14 genotypes of which 9 (64.3%) genotypes were from North Gondar, four (28.6%) from North Shewa and one (7.1%) from Arsi-Bale. Sub-cluster 4 contained 25 genotypes of which two genotypes (8%) were from East Gojjam 2, one genotype (4%) from North Gondar, 12 genotypes (48%) from Central Gondar, and 10 genotypes (40%) from North Shewa. Sub-cluster 5 included 9 (64.3%) genotypes from North Wollo, three (21.4%) genotypes from West Shewa, and two genotypes (14.3%) from Arsi-Bale. Sub-cluster 6 represented a heterogeneous group which constituted 33 genotypes of which two (6.1%) genotypes were from Central Gondar, 8 (24.2%) genotypes

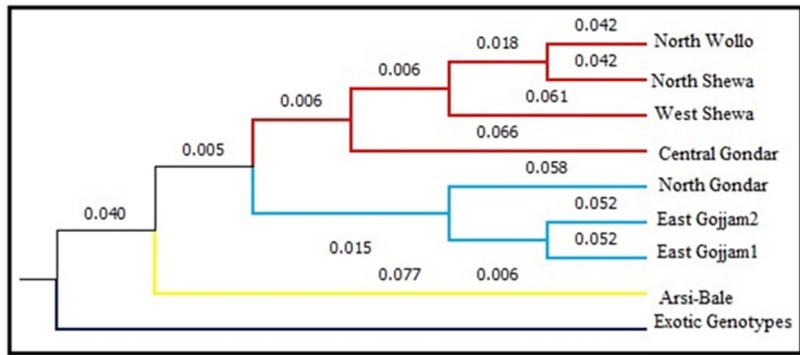

**Fig 4. UPGMA dendrogram showing the genetic relationships of nine chickpea populations collection areas.**

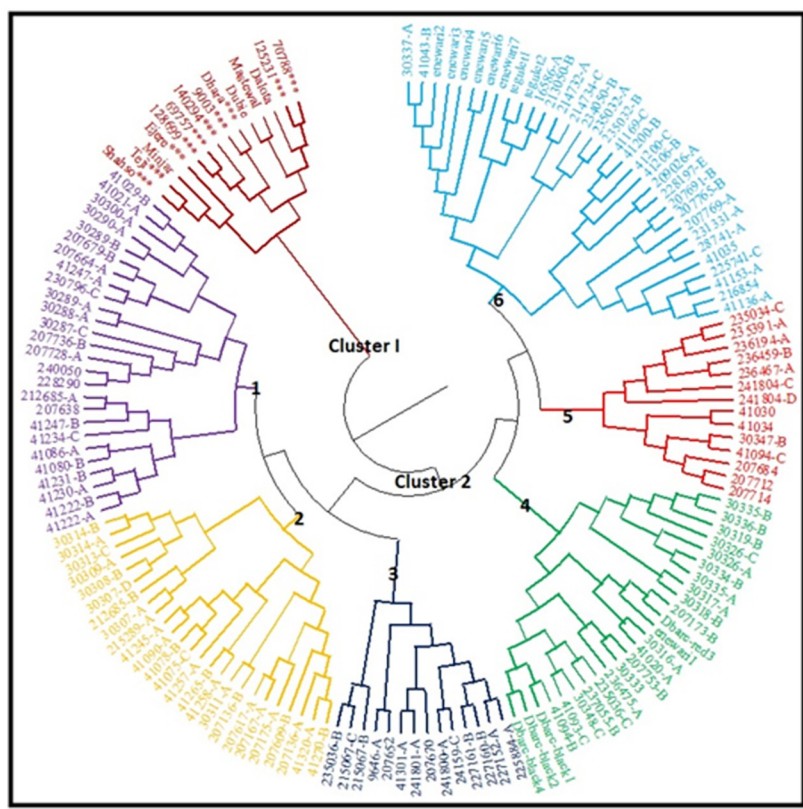

**Fig 5. UPGMA based dendrogram of 152 chickpea genotypes obtained using 23 SSR markers and Niel 1983 frequency based distance (\*\*\* Kabuli types chickpea, while the rest are desi type chickpea).**

from North Shewa, seven genotypes (21.2%) from North Wollo, nine (9%) genotype from West Shewa and seven genotypes (21.2%) from Arsi-Bale.

## Population structure

The population structure of the 152 chickpea genotypes was analyzed and the results showed that the highest peak was observed at K = 2 indicating the presence of two major clusters (Fig 6

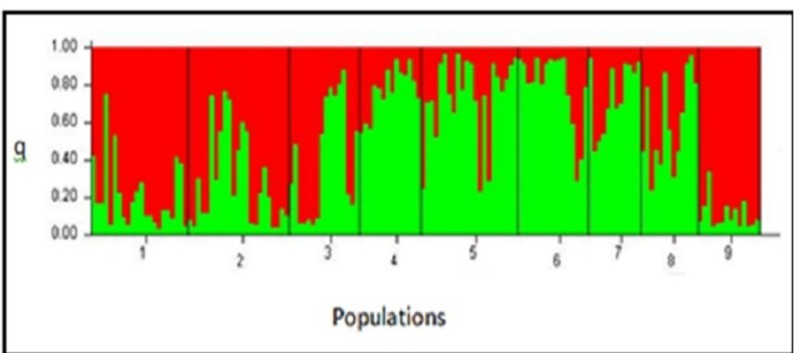

**Fig 6. Structure bar plot assigning 152 chickpea genotypes in two groups (K = 2) based on 23 SSR markers analyzed by the structure program, showing proportion of the two groups in each zones, where 1 = East Gojjam1, 2 = East Gojjam2, 3 = North Gondar, 4 = Central Gondar, 5 = North Shewa, 6 is North Wollo, 7 is West Shewa, 8 = Arsi-Bale and 9 = Exotic Genotypes, q = membership coefficient.**

**Table 8. Proportion of membership of each predefined nine population in each of the clusters obtained at the best K (k = 2).**

| Predefined Populations | Total Number of genotypes | Admixed | | Cluster 1 | | Cluster 2 | |
|---|---|---|---|---|---|---|---|
| | | Number of genotypes | Proportion in % | Number of genotypes | Proportion in % | Number of genotypes | Proportion in % |
| East Gojjam1 | 22 | 8 | 11.9 | 14 | 32.6 | 0 | 0.0 |
| East Gojjam2 | 23 | 13 | 19.4 | 10 | 23.3 | 2 | 4.8 |
| North Gondar | 16 | 7 | 10.4 | 6 | 14.0 | 3 | 7.1 |
| Central Gondar | 14 | 6 | 9.0 | 0 | 0.0 | 8 | 19.0 |
| North Shewa | 22 | 12 | 17.9 | 0 | 0.0 | 8 | 19.0 |
| North Wollo | 16 | 5 | 7.5 | 0 | 0.0 | 11 | 26.2 |
| West Shewa | 12 | 6 | 9.0 | 0 | 0.0 | 6 | 14.3 |
| Arsi-Bale | 13 | 9 | 13.4 | 0 | 0.0 | 4 | 9.5 |
| Exotic Genotypes | 14 | 1 | 1.5 | 13 | 30.2 | 0 | 0.0 |
| Total | 152 | 67 | 44.1 | 43 | 28.3 | 42 | 27.6 |
| Average Distance | - | - | - | 0.6565 | | 0.6408 | |
| Mean value of Fst | - | - | - | 0.0396 | | 0.0904 | |

Fst is genetic differentiation

and Table 8). The result from STRUCTURE analysis further confirmed results of the UPGMA tree clustering. Based on the probable likelihood of each genotype to be grouped into any of the two distinct groups, a total of 85 genotypes (55.9%) were grouped into one of the two populations. The first cluster which was of 43 (28.3% of total genotypes) genotypes were grouped into population 1, the next 42 (27.6%) into population 2. The remaining 67 genotypes (44.1%) were placed in the admixture group (Table 8). Cluster I was composed of 32.6%, 23.3%, 14% and 30.2% East Gojjam1, East Gojjam2, North Gondar and the exotic genotypes, respectively. Cluster II was made from 4.8%, 7.1%, 19.0 5, 19.0%, 26.2%, 14.3%, 9.5% from East Gojjam2, North Gondar, Central Gondar, North Shewa, North Wollo, West Shewa, Arsi-Bale, respectively. All population contributed to admixed group with variable proportion ranging from 1.5% (Exotic Genotypes) to 19.4% (East Gojjam2).

## Discussion

Efficient germplasm conservation and sustainable utilization requires a clear understanding of the genetic structure, diversity, and relationships among chickpea genotypes. This information is also helpful for breeders to identify new sources of germplasm harboring valuable alleles for improving yield, grain quality, and enhancing the level of resistance in cultivated varieties to various biotic and abiotic stresses [14, 50]. Molecular diversity and population structure studies using SSR markers for Ethiopian chickpea are limited. Therefore, this work was initiated with the main objective of analyzing the genetic structure, diversity, and relatedness of Ethiopian chickpeas genotypes, improved varieties, and exotic chickpea genotypes received from ICARDA using 23 SSR markers.

Result from SSR analysis indicated the presence of considerable allelic richness per locus, relatively moderate to high PIC, Ho and He values, and the presence of private alleles. High level of genetic diversity indicates the existence of molecular variation among the analyzed chickpea genotypes. High PIC values were also reported by Sefera et al [26], Getahun et al. [28] and Ghaffari et al. [51] which is in agreement with the present study, however a lower number of effective alleles per locus was recorded in the present study in contrast to that of Sefera et al [26] and Getahun et al [28]. This happened because of the different number of accessions,

different number of loci examined, and the nature of markers used in each study. However, comparable results were reported from Keneni et al. [25]. The high level of PIC values were an indicator of the efficiency of the markers for diversity studies in chickpea genotypes because a locus, with an estimated PIC value greater than 0.50, is considered to be highly diverse [52]. Nineteen markers had a score of 0.5 and above which indicates that these markers are highly informative SSR markers that could be employed in genetic diversity studies in chickpea. The ability of SSRs to detect intraspecific as well as interspecific variation in chickpea has been demonstrated by many authors [14, 28, 39, 53].

In this study, loci CaSTMS 11, CESSR 42, CESSRDB 54, GA 11, GA 24, SSR 22, and SSR 5 exhibited low-level of observed heterozygosity compared to the expected heterozygosity. Moreover, the high associated fixation index, implies that high levels of inbreeding among the assessed chickpea genotypes, which is expected because chickpea is a self-pollinated crop, previously only 0 to 1.58% of outcrossing was reported [51]. Simultaneously, loci CESSR 62 CESSR 71, NCPGR 45, NCPGR 53, NCPGR 94, SSR 1, SSR 4, TA 18, TR 1, TR 2, and TR 29 had a high-level of observed heterozygosity and low associated fixation index. This indicates that these loci could be associated with the occurrence of higher mutation rates or inbreeding depression [14]. The low level of heterozyogosity observed for the majority of the SSR markers are in accordance with other studies [14, 25]. However, higher level of heterozygosity was also reported for some SSR markers [24, 28, 54, 55]. According to Ghaffari et al. [51], allelic frequency <0.03 is considered as low, 0.03–0.20 considered as common, and > 0.20 considered as most frequent. Based on this delineation, rare alleles comprised 7.5% (10 alleles) of all the detected alleles while intermediate alleles accounted for 63.9% (85 alleles). The remaining alleles accounted for 28.6% of the allelic frequency (38 alleles) (data are not included).

All of the nine chickpea populations had a high percentage of polymorphism among the populations with the range of 95.7% to 100% and average of 99.5%. Comparable values of Shannon's Information Index were recorded for all populations. A relatively high number of alleles, effective alleles, and Shannon's information index were recorded in East Gojjam 2, which implies that chickpea genotypes from East Gojjam 2 are more diverse than the remaining chickpea collections of other geographic regions. The low-levels of private alleles were recorded in East Gojjam 2, North Shewa, North Wollo and Arsi Bale. Matus and Hayes [56] suggested that the occurrence of unique alleles could be an indication of the relatively high rate of mutation and diversity at SSR loci. The occurrence of unique or rare alleles has the potential to serve as a source of novel alleles for plant breeding and also provides an immense opportunity for generation of comprehensive fingerprint database for establishing genotype identity [57]. The percentage of polymorphism among Ethiopian chickpea populations discovered by Keneni et al. [25] and Getahun et al. [28] were lower than the present finding. The differences in values for estimated genetic diversity parameters between studies may be explained by different types and numbers of genotypes, different numbers and types of loci examined and perhaps the nature of markers used in each study.

AMOVA results indicate much of the variation was accounted for by the variation within population rather than among populations, suggesting that individual variation was more important for chickpea breeding programs. The low-level of molecular variation among population indicates that the presence of a high number of shared alleles among populations collected from different origins [58]. The exotic genotypes contributed 7.6% to the total molecular variation which provided an opportunity to expand the chickpea gene pool of Ethiopian origin, if there is no complete replacement of local germplasm with the improved ones. A low-level of molecular variation among chickpea populations were also reported from Keneni et al. [25] and Getahun et al. [28] for Ethiopian genotypes and Valadez-Moctezuma et al. [50] for Mexican chickpea. According to Wright [59] the combination of Fst rating, Fst value of

0.00 to 0.05 indicates low, 0.05–0.15 indicates moderate, 0.15–0.25 indicates high, and 0.25 indicates a very high-level of differentiation. Based on this delineation, the Fst score for the present study could be rated as low to moderate level of differentiation among populations with an increased level of admixtures which is the possible reason for the existence of the low-level of molecular variation among populations. Similar observation was made in cowpea [60]. The lower level of variation among populations might be attributed to germplasm exchange among regions and this is further confirmed from the result of pairwise gene flow (Nm) values among populations which were scored within the range of 1.12 to 4.87 exhibiting gene exchange among populations. A Nm value greater than 1 is considered an indicator of adequate gene flow among populations [61].

The genetic distance results showed that the genetic distance between each of the Ethiopian populations (eight populations) and the exotic population was higher than any pair of combinations within Ethiopian populations. This indicates that the genetic similarity between the exotic genotype and the Ethiopian populations is low, implying that Ethiopian populations are distantly related to exotic genotypes. However, close distance was estimated among Ethiopian populations collected from different regions, indicating that the highest genetic similarity was existed among Ethiopian chickpea genotypes. These results are in agreement with findings from Keneni et al. [25] and Getahun et al. [28]. In addition, UPGMA dendrogram tree of nine chickpea populations based on origins showed tendencies to be grouped together which indicates that the patterns of genetic relationships are among proximity areas of collections.

PCoA result indicates that the Ethiopian genotypes were uniformly distributed in the four quadrants regardless of their geographic origins, while the exotic genotypes were grouped in quadrant I forming sub-clusters which are distinct from the local genotypes. Genotypes from East Gojjam 2, North Shewa, and Arsi-Bale were highly diverse because they were evenly distributed in the three quadrants regardless of their geographic origins. However, some Ethiopian genotypes and the exotic genotypes appeared to follow geographic origins from which the genotypes were obtained. This result is supported by earlier studies using SSR markers [14, 25, 42]. The distinct identity of the exotic genotypes could be a consequence of deliberate selection criteria followed by the breeders in the development of these varieties [14].

The dendrogram tree constructed using the UPGMA clustering algorithm, clearly delineated the genotypes into two major clusters, Cluster I and Cluster II. Cluster II sub-divided into six sub-clusters, each consisted of variable number of genotypes. The exotic genotypes grouped in a single cluster. Results generated from dendrogram were also in agreement with those of the PCoA result. The patterns of genotypes clustering based on their geographic region were not consistent because some genotypes were grouped together according to their geographical proximity. This situation implies genetic distance doesn't follow geographical distance. Similar trends were reported by earlier works in chickpea [24, 28, 50]. The most probable reason could be seed exchange, and/or trade between farmers, leading to gene flow across boundaries within those areas. The dendrogram did not indicate any clear divisions between desi and kabuli type chickpea in the exotic genotypes. This may be due to the markers used for this experiment were not directly related with the characteristics that differentiate kabuli from desi type chickpea [50]. However, various authors have reported that the clustering of chickpea genotypes appears to follow geographic distribution from where these germplasm lines were obtained [41, 51, 53, 55] and Sefera et al. [26] and Getahun et al. [28] showed SSR markers in discriminating kabuli genotypes with that of desi genotype.

Applications of model-based clustering methods in the STRUCTURE software is helpful to demonstrate the presence of population structure, identify distinct genetic populations, assign individuals to populations, and identifies admixed individuals [47]. In the present study, a

structured population in chickpea was revealed, and was divided into two groups. The analysis of population structure revealed similarity with the results obtained from UPGMA clustering. The chickpea genotypes used for this study evolved from two population types showing varying degrees of introgression of the two types into respective genotypes. Structure is considered to be uniform when more than 80% of the accessions in one group have more than 80% membership of the group [14, 47]. There were no genotypes showing uniform structure with 100% membership in their cluster, indicating that the existence of gene flow or introgression was apparent. According to Gemechu et al. [25] result Ethiopian chickpea germplasm of different collection of this study were grouped into five clusters of distinct genetic populations. They proposed that the genotypes resulted from independent evolutionary mechanisms (genetic drift, mutation, migration, selection, and in flux/out flux of genes in the form of germplasm exchange) that split them into discrete gene pools. Gene introgression is critical for breeders for variety development programs because it provides essential trait combinations such as improved agronomic features, high resilience to environmental challenges, diseases, and insects, as well as other benefits such as improved nutritional quality [14]. It is also applicable to broaden the genetic base of chickpea genotypes through crossing programs.

## Conclusions

The magnitude and pattern of genetic variation was estimated, which indicated that a considerable genetic diversity existed in Ethiopia chickpea genotypes. The results also further confirmed the efficiency and effectiveness of SSR markers to study genetic diversity in chickpea. This result will have a direct applicability for efficient and systematic conservation and sustainable utilization of germplasm. This result can assist chickpea breeders in selecting diverse parental materials for crossing activities to take the advantage of heterosis value. The results are also helpful for genebank managers because there are large numbers of genotypes clustering in one group collected from the same locality and it seems that these genotypes are duplicated genotypes which are the major problems in germplasm conservations. To reduce the high amount of redundancy in germplasm collections, techniques including deliberate bulking and the establishment of core collections must be implemented. Though this work provided preliminary information regarding the existences of genetic diversity, studies related to marker traits association are required. Therefore, a comprehensive study to map the associations of the markers with agronomic traits of economic importance is required.

## Supporting information

**S1 Text. Passport data for 152 chickpea genotypes used for the study.**
(XLSX)

## Acknowledgments

The first author wishes to acknowledge Ethiopian Biodiversity Institute for giving him study leave and Addis Ababa University for material and technical supports to this study as a component of his PhD thesis. Ethiopian Biodiversity Institute, the International Center for Agricultural Research in the Dry Areas (ICARDA) and the National Chickpea Research Project in Ethiopia for providing chickpea genotypes for the study. Moreover, the study was a collaborative effort with a professional from the United States Department of Agriculture, which is an equal opportunity provider and employer.

## Author Contributions

**Conceptualization:** Sintayehu Admas, Kassahun Tesfaye, Teklehaimanot Haileselassie, Eleni Shiferaw, K. Colton Flynn.

**Data curation:** Sintayehu Admas.

**Formal analysis:** Sintayehu Admas, Eleni Shiferaw.

**Funding acquisition:** Eleni Shiferaw, K. Colton Flynn.

**Investigation:** Sintayehu Admas, Kassahun Tesfaye, Teklehaimanot Haileselassie.

**Methodology:** Sintayehu Admas, Kassahun Tesfaye, Teklehaimanot Haileselassie.

**Project administration:** Kassahun Tesfaye, Teklehaimanot Haileselassie.

**Resources:** Kassahun Tesfaye, Teklehaimanot Haileselassie, Eleni Shiferaw, K. Colton Flynn.

**Software:** Eleni Shiferaw.

**Supervision:** Kassahun Tesfaye, Teklehaimanot Haileselassie, Eleni Shiferaw, K. Colton Flynn.

**Validation:** Eleni Shiferaw.

**Visualization:** Kassahun Tesfaye, Teklehaimanot Haileselassie, Eleni Shiferaw.

**Writing – original draft:** Sintayehu Admas.

**Writing – review & editing:** Kassahun Tesfaye, Teklehaimanot Haileselassie, Eleni Shiferaw, K. Colton Flynn.

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
