## [Decision Letter · Decision Letter 0]

19 Aug 2021

PONE-D-21-21598

Genetic variability and Population Structure of Ethiopian chickpea (Cicer arietinum L.) germplasm

PLOS ONE

Dear Dr. Admas,

Thank you for submitting your manuscript to PLOS ONE. After careful consideration, we feel that it has merit but does not fully meet PLOS ONE’s publication criteria as it currently stands. Therefore, we invite you to submit a revised version of the manuscript that addresses the points raised during the review process.

We look forward to receiving your revised manuscript.

Kind regards,

Roberto Papa, PhD

Academic Editor

PLOS ONE

Journal Requirements:

“The study is part of the first author PhD thesis funded by Ethiopia Biodiversity Institute and Addis Ababa University.”

Reviewers' comments:

Reviewer's Responses to Questions

**Comments to the Author**

1. Is the manuscript technically sound, and do the data support the conclusions?

Reviewer #1: Yes

Reviewer #2: Yes

2. Has the statistical analysis been performed appropriately and rigorously? 

Reviewer #1: Yes

Reviewer #2: Yes

3. Have the authors made all data underlying the findings in their manuscript fully available?

Reviewer #1: Yes

Reviewer #2: Yes

4. Is the manuscript presented in an intelligible fashion and written in standard English?

Reviewer #1: Yes

Reviewer #2: Yes

5. Review Comments to the Author

Reviewer #1: In general I've found that the article is well written, however I did not find that results are particularly interesting, simply there are no great differences between accessions based on geographical origin. Despite could have been an interesting point to understand wheter the geography of Ethiopia shaped the genetic background of chickpea ethiopian population, i would suggest to include not only released variaties and breeding lines but eventually to concentrate also on landraces, in order to understand if the gene flow and introgression happend also within local varieties, which could be an interesting point. Moreover I would like to see more clearly the geograohical distribution of accessions considering it is what is defining your groups, indeed I can't take an ethiopian map and control the place of each district you cite, I would add a map, eventually to the supplementary. Moreover, majority of studies on genetic resources in ethiopia take into account the aspect of altitude, which is an interesting point in ethiopian landscape, you could try to group districts by altitude and see wheter this clusterize better your accessions but as curiosity. However the statistical analysis you have done are completed and well executed and everything is well explained. I would absolutely reccomend you to redo the image of gel electophoresis from zero as it is not suited for publication. I also want to see images in a better resolution for publication, here and there is impossible to distinguish the names of accessions. I do believe that your conclusions could however be useful for future breeding program, however I still have doubts on the choice of the vegetal material for this kind of study. Finally, without any doubt the population structure result is the more interesting, partly confirmed by other analysis, thus I would discuss it deeper in discussion and eventually hypothesize the reasons of such a structuring. I would also being interested in a more precise characterization of the genetic resources you used, for example the habitus, the group (desi or kabuli), something easy that could help the reader understanding with which type of material we are dealing with.

However the work is well done, I appreciate the writing which is clear and well summarized. English is good and in general it is ok for publication (apart from images resolution that you have to correct).

Reviewer #2: overall the work seems well done, I have no reservations regarding the analyzes carried out, and the manuscript is written in correct English.

however, I have some suggestions to present.

minor issues

I suggest some adjustments in the form:

- from lines 65 to 69: the sentence is long and not very fluent.

- from lines 79 to 83: shorter and more fluently.

- from lines 448 to 451: the sentence is long and not very fluent.

major issues

- in the materials and methods, it is not specified if the 138 genotypes are landraces, explain better

- in the discussions the author writes: "The dendrogram did not indicate any clear divisions between desi and kabuli type chickpea ",

but the material distinction in desi and kabuli has not been addressed in materials and methods.

therefore

overall I suggest a better description of the material used, moreover, to insert a figure of the geographical distribution of the material used.

-I suggest a better quality of results of the images used, in particular Fig1 and Fig3.

-Finally, I suggest the author to spend a few lines to comment on a possible reason in the population structure for the presence of two groups.

6. PLOS authors have the option to publish the peer review history of their article (what does this mean?). If published, this will include your full peer review and any attached files.

Reviewer #1: No

Reviewer #2: No

---

## [Author Response · Author response to Decision Letter 0]

27 Aug 2021

I have included in the revised manuscript all comments and suggestions given by reviewers. Except the following points

1. Clustering genotypes based on altitude: The result did not show altitudinal clustering patterns indicating that there were no great differences between accessions based on altitudinal gradients. Because of the study was targeted at the major chickpea growing areas of Ethiopia. The altitudinal variations among the collection sites were narrow, which makes difficulty to see altitudinal effects to the genotypes. In addition, the informal seed exchange systems have been practiced for centuries among traditional farmers in Ethiopia. So, we got similar results that of geographical origin and we did not present the result to limit the size of the paper. (For Reviewer 1) 

2. Fig 1 (now Fig 2): is not addressed because silver stained gel picture have most of the time less resolution power. (Both Reviewers)

---

## [Decision Letter · Decision Letter 1]

15 Nov 2021

Genetic variability and population structure of Ethiopian chickpea (Cicer arietinum L.) germplasm

PONE-D-21-21598R1

Dear Dr. Admas,

We’re pleased to inform you that your manuscript has been judged scientifically suitable for publication and will be formally accepted for publication once it meets all outstanding technical requirements.

Kind regards,

Roberto Papa, PhD

Academic Editor

PLOS ONE

Additional Editor Comments (optional):

Reviewers' comments:

Reviewer's Responses to Questions

**Comments to the Author**

1. If the authors have adequately addressed your comments raised in a previous round of review and you feel that this manuscript is now acceptable for publication, you may indicate that here to bypass the “Comments to the Author” section, enter your conflict of interest statement in the “Confidential to Editor” section, and submit your "Accept" recommendation.

Reviewer #1: All comments have been addressed

Reviewer #2: All comments have been addressed

2. Is the manuscript technically sound, and do the data support the conclusions?

Reviewer #1: Yes

Reviewer #2: Yes

3. Has the statistical analysis been performed appropriately and rigorously? 

Reviewer #1: Yes

Reviewer #2: Yes

4. Have the authors made all data underlying the findings in their manuscript fully available?

Reviewer #1: Yes

Reviewer #2: Yes

5. Is the manuscript presented in an intelligible fashion and written in standard English?

Reviewer #1: Yes

Reviewer #2: Yes

6. Review Comments to the Author

Reviewer #1: I have really appreciated that authors added a description of the vegetal material used in the research and also the map showing the geographical points of collection is interesting and I'm glad they haved decided to include it.

In my opinion the written parts added to the text and the corrections made by the author lead to a clearer and more fluent reading of the article.

Reviewer #2: I believe that the revisions and changes are sufficient, making the paper complete and more fluent in content and written form.

7. PLOS authors have the option to publish the peer review history of their article (what does this mean?). If published, this will include your full peer review and any attached files.

Reviewer #1: No

Reviewer #2: No

---

## [Editor Report · Acceptance letter]

16 Nov 2021

PONE-D-21-21598R1 

Genetic variability and population structure of Ethiopian chickpea (*Cicer arietinum * L.) germplasm 

Dear Dr. Admas:

I'm pleased to inform you that your manuscript has been deemed suitable for publication in PLOS ONE. Congratulations! Your manuscript is now with our production department. 

Kind regards, 

on behalf of

Prof. Roberto Papa 

Academic Editor

PLOS ONE